# Evaluation of Medical Staff Satisfaction for Workplace Architecture in Temporary COVID-19 Hospital: A Case Study in Gdańsk, Poland

**DOI:** 10.3390/ijerph20010639

**Published:** 2022-12-30

**Authors:** Agnieszka Gebczynska-Janowicz, Rafal Janowicz, Wojciech Targowski, Rafal Cudnik, Krystyna Paszko, Karolina M. Zielinska-Dabkowska

**Affiliations:** 1Faculty of Architecture, Gdańsk University of Technology, 80-233 Gdansk, Poland; 2Copernicus Podmiot Leczniczy Sp. z o. o., 80-803 Gdansk, Poland; 3Institute of Nursing and Midwifery, Medical University of Gdańsk, 80-210 Gdansk, Poland

**Keywords:** healthcare architecture, workplace architecture, intensive care unit, ICU nurses and COVID-19 pandemic, temporary hospital, adaptive reuse, COVID-19 hospital

## Abstract

This article analyses the architecture that was used in the temporary AmberExpo hospital in Gdańsk, Poland which was installed during the COVID-19 pandemic. The construction of this type of facility is often based on experimental approaches, aimed at caring for patients suffering from an infectious disease in emergency conditions. In order to assess the level of employee satisfaction with the architectural and technical elements used in the first period of the hospital’s activity, medical staff were asked to fill out a questionnaire. The analysis of the survey’s results indicated that the majority of employees expressed satisfaction with the architectural and technical elements, with the design of the spatial layout of the individual medical zones receiving the most positive feedback. However, frequently selected drawbacks in the design included the lack of natural daylight, the artificial light that was used and the acoustics of the facility. This detailed examination of the satisfaction and feedback from medical employees working in this type of emergency facility enables the development of solutions that in the future will allow for the improved adaptive reuse and implementation of such structures, with enhanced time and economic efficiency, and most importantly, the ability to provide a safer workplace.

## 1. Introduction

The rather stable political, economic and epidemiological global situation at the turn of the 20th and 21st centuries led to a situation in which most countries could not find any rationale behind investing in emergency buildings that could protect citizens in the event of war, natural disasters, or a pandemic threat. Indeed, the urgent need to approach the issue of medical reserves that would be able to relieve the basic healthcare system in the case of unusual rescue operations was only addressed in 2020 by, among others, scientists from Sweden shortly before the pandemic was announced [1]. The history of the COVID-19 pandemic demonstrates that a significant portion of the contemporary healthcare system was not organisationally or infrastructurally ready for the increased number of patients generated by the rapid growth in the number of infections.

Medical-function buildings are complex structures where architecture is harmonised with advanced building systems. Their spatial arrangement plan includes the requirements imposed by special medical equipment and procedures that ensure operational safety for patients and staff [2,3,4,5]. Significant investment outlay followed by high operational costs means that medical buildings are rarely designed with a reserve in the functional and spatial program that would enable their operation in an emergency wherein a sudden increase in the demand for medical services comes about.

Severe acute respiratory syndrome (SARS), a viral respiratory disease, was diagnosed for the first time in February 2003 in China. The SARS-CoV epidemic spread to 29 countries and regions. Disease transmission through direct contact of the sick and healthy have been documented in, among others, Toronto, Canada; Hong Kong; Singapore and Hanoi, Vietnam. In July, WHO announced the SARS pandemic was over [6].

Experience related to overcoming and mitigating the outcomes of the pandemic forced research on the possibilities to develop organisational structures within the healthcare system that would constitute backup facilities in an emergency. Numerous researchers suggested adapting medical units to the possible occurrence of a new pandemic [7]. One of the most important objectives of the research commenced at the time involved organisational and spatial structures of hospitals and outpatient clinics since analysing the effects of the SARS pandemic of 2003 indicated that its characteristic feature was a high level of infections recorded among medical professionals due to nosocomial transmission. On a global scale, 21% of all SARS cases were people working in medical facilities [6]. This led to taking actions aimed at developing an emergency case response strategy for events associated with an infectious disease pandemic [8,9]. Some of the studies regarding epidemiological issues point to the role of architectural and engineering factors in preventing the spread of viral and bacterial infections within the physical environment of a hospital [10,11]. 

The results of a study on the role of medical unit architecture in reducing infection transmissions turned out to be useful already in 2019. A growing number of cases of a new virus known as severe acute respiratory syndrome coronavirus 2 (SARS-CoV-2) were reported in December. The virus brought about acute viral pneumonia in Wuhan, China. The global spread was extremely rapid and affected major population clusters on most continents. On 11 March 2020, WHO announced it to be a global pandemic since the number of cases and fatalities grew worldwide [12] (at the time of writing this article, the world is still battling its outcomes). Individual countries were thus compelled to adapt the capabilities of their healthcare system to the increasing number of patients who require medical attention. At the time of writing this article, in November 2022, WHO reports indicated an increase in the number of infections and unpredictable disease development scenarios associated with the appearing mutations [13].

The health system burdened by the pandemic required the construction of emergency medical buildings, e.g., temporary hospitals. An important type of temporary hospital, the model of which was developed during the COVID-19 pandemic, was the adaptation of large-scale facilities, such as an exhibition or concert hall, for the needs of a medical facility. The first units of this type were created in Wuhan on 5 February 2020. The wave of infections reached a critical level at that time with thousands of new cases per day; therefore, the city launched three facilities called Fangcang. Fangcang hospitals were large-scale, temporary hospitals quickly constructed by converting existing public facilities, such as stadiums and exhibition centres, into healthcare facilities. They were primarily used to isolate patients with mild-to-moderate COVID-19 symptoms. Other fundamental functions of the rooms created therein include patient segregation, basic medical care, patient health condition monitoring and a quick referral to other advanced medical facilities in a life-threatening situation. Adapting service facilities included changing interior fit-out and installing beds, medical devices and materials supporting medical care [14]. By isolating and treating only mild-to-moderate cases, Fangcang hospitals in Wuhan released the limited medical infrastructure of higher-tier hospitals, such as wards ensuring respiratory support and intensive care for patients with severe COVID-19 symptoms.

Experiences from Wuhan associated with the functioning of Fangcang hospitals have become an impetus to search for successive architectural solutions. Building models that are originally used for recreation or commercial purposes, among others, sports venues or exhibition centres were converted into medical facilities [15]. The large volume of such buildings allows for observing the development of the disease in a larger number of patients and conducting tests supporting the fight against its spread [16].

## 2. Materials and Methods

### 2.1. Study Location

The case study for the conducted analysis was the architecture of the temporary AmberExpo hospital created in Poland at the Gdansk International Fair as support for the healthcare system infrastructure in terms of actions taken in association with the COVID-19 pandemic within the Pomorskie province. The existing exhibition building was adapted to the needs of a temporary medical unit.

Information on the architecture and the associated infrastructure of the constructed temporary AmberExpo hospital was obtained from the administrator of the Gdansk International Fair (MTG) building. The data on its functioning originates from the reports by the Provincial Epidemiological Station in Gdansk and the medical facility manager. The photographic material was acquired in February 2021, prior to commissioning the facility, and in June during its operation within its first activation period.

To assess the impact of adopted architectonic and technical solutions on the work environment, the authors conducted a questionnaire among the medical professionals working within the hospital in question during the COVID-19 pandemic. 

### 2.2. Study Design

A questionnaire was conducted during the pandemic. It was aimed at assessing the level of temporary hospital workplace employee satisfaction as well as the architectural solutions that contributed to it.

### 2.3. Participants and Setting

During the pandemic, in total, 123 doctors, 233 nurses, 85 paramedics, 10 X-ray technicians/radiologists, 10 front desk receptionists and 200 nursing assistants worked in the temporary hospital. The questionnaire was sent to 100 healthcare professionals working in the temporary hospital in Gdansk during its first period of activity from March to June 2021.

The questionnaire was comprised of two parts. The first part contained three questions concerning the demographic characteristics of the participants taking part in the study. Specifically, these questions concerned gender, age and healthcare profession of these people. (Table 1). Accordingly, 9 (28.1%) people out of the 32 employees participating in the questionnaire were males, and 23 (71.9%) were women. The respondents were aged 50–60+ (9 people–28.1%); 30–49 (16 people–50%) and 18–29 (7 people–21%). The temporary hospital personnel that replied to the questionnaire involved such professional groups as hospital nurses (56.3%), hospital physicians (28.1%), cleaning personnel (3.1%) and Technical Support (12.5%).

The second part contained 10 questions concerning various aspects of the assessment of satisfaction with architectural and technical solutions adopted at the AmberExpo hospital. Two of them were multiple-choice, closed-end questions while the other elements use the answer formula according to the Likert scale—the format of utterances arranged in order from total acceptance to total rejection.

Data collection began on 9 June 2021 and ended on 1 September 2021.

### 2.4. Data Analysis

Personnel satisfaction was measured using a 13-point self-completed questionnaire, developed using Google Forms: Free Online Surveys for Personal Use. It was conducted with the consent of MTG administrators. The analysis was based on descriptive statistics.

## 3. Results

### 3.1. General Characteristics of the Study Sites

The emergency temporary hospital was implemented at the AmberExpo exhibition hall in Gdansk, which acted as an emergency medical facility for several months and constituted an expansion of the existing medical facility—the Copernicus Podmiot Leczniczy hospital. The Ordering Party was the Voivode of the Pomorskie province, a representative of the Polish government.

The design team that created the concept for the AmberExpo temporary hospital consisted of the following companies: FORT TARGOWSKI Sp. z o.o. (architecture), Wepa Projekt s.c. Pachnik S. Welenc A. (mechanical ventilation systems) and AS Pracownia Projektowa A. Szypowicz (electrical installation).

The functional layout of this type of field hospital was adapted to the needs of handling COVID-19 patients who required nursing care as well as part of the intensive care unit. It was assumed that it would accept patients infected with the SARS CoV-2 virus while the diagnostics in terms of their qualification would be conducted at an earlier stage.

The objective of the investment project was to adapt an exhibition building to function as a hybrid facility, which was by default used as an exhibition building but also ready for quick transformation into a medical facility in the event of a mass-casualty incident. The assumption of the investor was to employ funds not only on the immediate needs of the medical facilities constituting an expansion of the medical units during the COVID-19 pandemic but also as an investment for a possible reactivation in a hypothetical emergency situation. At the facility planning stage, it was assumed to fully utilise the existing physical space of the building, supplemented with temporary elements required for a medical unit, and the fixed elements, constituting the adaptation to a hybrid building function. It was also assumed that the main objective of the investment is the possible reuse of introduced fixed elements in the event of a mass-casualty incident and the operational safety of such a space.

The subject matter of the adaptation for the purpose of a temporary hospital was an area consisting of several buildings of the Centrum Wystawienniczo–Targowe MTG SA at 11 Żaglowa Street in Gdansk: exhibition halls A, B and C and the first two floors of the administrative buildings. The adapted trade fair complex is a medium-height building, connected with a single-floor exhibition area of 11,947.51 m^2^, divided into three sections (Figure 1).

The AmberExpo temporary hospital was commissioned in February 2021 as a facility adapted for the next wave of COVID-19 infections among the residents of the Pomorskie province. Accordingly, 28 primary care beds and 10 stations adapted for anaesthesiology and intensive care procedures were launched on 8 March 2021. Two weeks later, the number of primary care beds in the temporary hospital was increased to 122. The facility was adapted to ultimately have 380 beds and 18 intensive care stations installed. It terminated its activities on 30 June 2021. During its 4-month operation, it admitted 808 patients, 90 of whom were treated at the intensive care unit. Patients spent 11 days on average in the hospital.

In the months from July to November 2021, the building acted as an exhibition centre organising mass trade fair events. In light of the growing number of COVID-19 cases in Poland and the increasing burden on the healthcare system in the Pomorskie province, the authorities started to consider reactivating the temporary hospital in order to relieve stationary hospitals and, above all, to ensure an internal medical hospital bed reserve for the province. On 14 December 2021, the Minister of Health decided to reopen the temporary hospital. The first patients were admitted a month later.

The architectural design for the MTG building adaptation to a temporary hospital assumed the need to integrate organisational and architectural activities in order to effectively implement preventive actions against the spread of infectious diseases. When defining the areas, the designers used existing physical barriers, which, after modification, enabled the organisation of clean and dirty sluice room sets to ensure the possibility of properly entering and exiting infectious zones and the safety of movement between clean and dirty zones for the personnel. The organisational diagram was based on planning places where specific procedures and activities were to be conducted. This resulted in separating the passageways of patients, personnel, clean materials introduced into the facility and dirty materials evacuated from the unit and intended for, e.g., disposal or sterilization (Figure 2). It also enabled obtaining a preassumed room containment class.

The functional layout of the temporary hospital can be divided into three main functional, interconnected zones: patient care rooms, rooms for medical personnel and a zone of supply and storage of materials necessary for the provision of medical services. It is necessary that each zone should be separated, and communication between them should take place using airlocks and entrances protected by access control. The location of the auxiliary zones should, on one hand, be outside the patient care area but on the other hand, in the immediate vicinity so as not to cause the need for redundant work resulting from the movement of personnel and materials between clean and contaminated areas.

A spatial plan in the scope of basic functionalities was adopted for the crisis associated with the COVID-19 pandemic and the adaptation of MTG buildings for the purposes of a temporary hospital (Figure 3):

Ground floor:·Casualty ward with additional immediate therapy beds;·Five nursing areas with a total capacity of 190 hospital beds;·Anaesthesiology and intensive care unit with a total capacity of 20 hospital beds;·Imaging diagnostics area equipped with mobile X-ray machines and a CAT scan;·Area for evacuating materials (sluice rooms for materials to be disposed of, sluice room for materials to be sterilised, sluice rooms acting as a mortuary);·Service and facility maintenance personnel area;·Hospital bed washing station room;·Area for collecting laboratory test samples;·Area with sluice rooms for introduced materials;·Hospital pharmacy department.

First floor:·Administrative and medical personnel leisure area;·Information and third-party handling area.

In architectural and organisational terms, the presence of the SARS-COV-2 pathogen made it necessary to analyse hospital structure with regard to the possible transmission of the infection onto other people as well as to study various transmission routes, including through contact, droplets and inhalation. In order to ensure an epidemiologically safe space, safety analyses were conducted based on the developed scenarios of possible infection transmission. This led to dividing the building into two sections: (1) the dirty part dedicated to infected patients (with a defined containment degree) and (2) the clean part dedicated to employees. Area-specific communication devices were adopted as a standard for the isolated area so as not to cause the need to carry them between the clean and dirty parts.

At the stage of commencing the development of the functional arrangement within the area infected with SARS CoV-2, a decision was made to employ various partitions within the area depending on sanitary requirements and their reuse potential. The main container development systems were used for the temporary system, e.g., sanitary and hygienic purposes (Figure 4a,b). Some locations, due to the analysis of the remobilization potential, involved separating new rooms that permanently adapted the building to a hybrid building function, e.g., sluice rooms using plasterboard partition walls or aluminium walls. Nursing area organisation was ensured through the application of solutions that enabled dividing the hall space by using temporary, free-standing trade fair partitions available within the facility. The use of these modular systems resulted from material availability within the facility and the flexibility and multifunctionality of the space in the trade fair buildings, that by its nature retains the ability to reorganise the space depending on the crisis development scenario or to restore the trade fair function (Figure 4c,d).

The facility distinguished dirty and clean routes as well as functional routes, which resulted from the conducted medical procedures. At the stage of assuming design solutions, all area-connecting routes were defined with simultaneous clear separation and marking of isolated areas. The adopted principle was aimed at ensuring effective management of the facility and limiting infection transmission. For this purpose, infected patient areas were isolated, and personnel areas were separated by sluice rooms with introduced systems to control access to the rooms, and a clear signage system, thus limiting the possibility of cross-infections. This enabled the introduction of two personnel requirements in terms of protective clothing for the isolated area and the administrative area. Contemporary research confirms that properly designed and implemented access zones and material, personnel and patient flows between such zones enable reducing nosocomial infection rates [9]. Moreover, the results of preliminary studies involving correctly planned areas of medical units operating within the COVID-19 regime enable ensuring epidemiological safety for their personnel [17]. Hence, a properly organised functional arrangement, supported by organisation activities, including personnel and patient training, enables controlling the risk of nosocomial infections due to these measures. This outcome was observed, among others, in the Fangcang shelter hospitals in China where the risk of internal nosocomial infections was not higher than in traditional hospitals [14].

A significant challenge in adapting the MTG was equipping the temporary hospital with the technical infrastructure required for a medical facility. This included mechanical ventilation, heating, medical gas system, electricity and CCTV. Some of the systems were modernised while some were designed from the beginning.

### 3.2. Questionnaire-Based Survey Regarding the Assessment of Spatial Solutions within the AmberExpo Hospital (Functional and Spatial Layout of the Architecture and the Technical Infrastructure)

Despite the initial concerns that the satisfaction of healthcare professionals working at a temporary facility would be low, the study results indicate that most of the surveyed people positively rated working at the AmberExpo temporary hospital. The question: “Would you say that the technical and organisational solutions applied in the MTG temporary hospitals made your work environment within the facility safe?” was positively answered by 93.8% of the respondents (Table 2).

The spatial arrangement was assessed very positively. Only one person poorly rated the functional and spatial solutions in the MTG temporary hospital (staff/material sluice rooms, patient zone partitions) providing the personnel with sanitary and hygienic safety. A total of 56.3% of the employees indicated spatial arrangement as the element that was organised better than in the facilities they worked within previously. Such a good result was influenced by clearly boxed-off sanitary zones, wider communication routes and large distances between patient beds.

Distances between the wards were a great functional hindrance. A pneumatic tube mail system was planned in anticipation of the large distance to be covered. A question related to this aspect was included in the questionnaire: “How would you rate the need for a pneumatic tube system within the MTG temporary hospital?” A total of 90.6% of the surveyed people rated this positively.

Depending on the development of COVID-19, temporary hospital patients are characterised by varying levels of fitness. A rest and refreshment area was provided for patients who were able to move by themselves. This area enables one to independently prepare a warm drink or eat a meal at a table. This solution was appreciated by 62.5% of all employees.

A fundamental disadvantage of an exhibition hall as a facility intended for emergency adaptation into a medical unit is the lack of sufficient natural lighting. Sunlight penetrates MTG exhibition halls through skylights. Despite using artificial lighting, this was the missing element that the respondent pointed to the most. A total of 59.4% of the employees responding to the questionnaire indicated the lack of daylight as the main aspect of worse conditions than in other medical facilities they worked at.

When comparing different areas of care for patients infected or suspected of being infected with SARS-CoV-2, the personnel believed this solution to be better than these applied in other medical units, especially in terms of medical equipment (68.8%) and spatial layout and communication between units within the facility (56.3%) (Table 3).

In their answer to the question “Which elements of the technical infrastructure at the MTG temporary hospital would you rate as a worse solution than that used in other medical facilities that you worked at?”, most respondents (59.4%) indicated artificial lighting. When adapting trade fair facilities, as was the case with the AmberExpo hospital, the original function determined hard-to-solve deficits, resulting from, among others, limited access to daylight and acoustic requirements. Consequently, arrangements with supervising authorities and a questionnaire study on the impact of technical solutions on personnel performance were conducted in the case of the Centrum Wystawienniczo–Targowe MTG SA acoustics (46.9%).

Due to an understanding of the significant impact of a temporary hospital’s architectural environment on the satisfaction of healthcare professionals, it is possible to make decisions on factors to be improved in order to increase the quality of temporary medical buildings to the highest level possible. The results of this study are significant in terms of creating other temporary hospital types.

## 4. Discussion

Attempts at quickly adapting to a critical situation associated with the pandemic are often short-term experiments. A significant portion of adaptive reuse examples includes expensive investments that cannot be applied again. Indeed, some of the field hospitals constructed hastily have not been fully utilised. However, such experiences are very valuable since they provide new data on the scope of a healthcare system’s readiness to operate in emergency situations and the adaptability of existing architecture to rapid changes [18].

Very few scientific publications that could support architects and designers in the creation of such units over a short time were available before 2020. In the reports on the activity of similar facilities, numerous scientists call for further analysis in this field and indicate a need for studies redefining field hospital architecture [19,20].

Working in this type of hospital is a big mental and physical burden for medical staff. Acting in a crisis situation, transferring to a new medical unit and contact with a disease, whose characteristics and complications are not clearly defined, were challenges even for experienced nurses.

During the three years of the pandemic, many studies were prepared to describe the experience of medical personnel in the fight against COVID-19 in various countries and various medical units [19,21]. The results of this ongoing research have shown that working during the pandemic has had a very large impact on the personal and professional lives of nurses, as they are one of the most exposed professional groups due to direct contact with patients. A high percentage of infections among nurses is indicated by the results of studies presented in the references based on the experiences of employees of individual medical units [22,23,24].

The personnel working on the front line of the fight against the pandemic were very heavily burdened. Working under crisis conditions caused mental health problems, such as fear, depression and insomnia [25,26]. This condition was caused mainly by the fear of disease, contact with the patients isolated from their families and care for their own families. As aggravating factors, nurses also distinguished the avalanche of duties, long-term fatigue and experiencing the death of patients [27,28,29]. In addition, nurses pointed to the physical overload related to the difficulty of working in protective clothing and the work overload due to staff shortages and equipment shortages. The need to quickly adapt to new conditions in experimental medical units in which there were unfavourable spatial solutions, such as short distances between beds, long distances between the individual functional zones and the lack of space for rest, were also indicated [28,30].

The difficult working environment for COVID-19 patients is already compared in some studies to the conditions prevailing in intensive care units [27], which, according to the conducted analyses, are considered to be the reason for the occurrence of professional burnout among nurses working there [31,32]. Burnout is a threat to the intensive care profession, which is why it is very important to reduce harmful factors in the work environment [33]. Such possibilities are offered by the architecture that, through the appropriate organisation of the space of the temporary hospital, is able to reduce the occurrence of hospital-acquired infections [34,35] and improve the well-being of medical personnel [36,37,38].

## 5. Conclusions

The conducted surveys allowed for the analysis of the architectural and technical solutions used in the AmberExpo hospital from the perspective of the users, the medical staff working there and the nurses. The results of the research indicated several factors shaping the workspace that directly affect the well-being and safety of the users: the arrangement of rooms ensuring epidemiological safety, the proper lighting of workstations, the equipment reducing the discomfort of work in a temporary hospital (the cameras at the patients’ beds, the pneumatic mail, etc.) and the properly planned social facilities system. The experience of the nurses may be useful in planning a safe healthcare system in the face of the possible occurrence of further infectious pandemics, as indicated by the conclusions of studies conducted by other researchers [39].

The analysis of the adaptation of the construction structure of the exhibition building to the needs of the medical facility made it possible to distinguish the advantages and disadvantages of such a solution. The building described in the article, thanks to the flexibility of the functional system and large area, enables the location of numerous hospital beds and the optimal distance between the beds, reducing epidemiological risks. Most technical infrastructure, such as mechanical ventilation, electrical network, industrial cameras and fire protection, in this case only need to be redeveloped and adapted to the characteristics of the medical facility. The addition of new infrastructures, such as medical gases, is made possible by the technical tunnels located under the floor. The experience of other temporary hospital investment projects shows that the presence of technical tunnels is a key asset of a building intended for the temporary performance of medical functions [40]. These aspects mean that in a short time, it is possible to plan working conditions similar to those of standard hospitals.

A significant disadvantage of creating a temporary hospital in an exhibition building is the lack of or insufficient access to natural light provided to workstations. Previous research revealed that natural light (daylight) very positively impacts health and well-being [41] and is of great significance in patient recovery and their contact with the outside world and the day/night cycle change [42]. In contrast, poorly designed artificial night lighting for hospital employees and patients can have an adverse effect [43]. In the case of the patient ward, the light only came from skylights, which is limited and functionally insufficient. For this reason, it was necessary to additionally illuminate the interior of the facility using artificial LED lighting in the form of suspended luminaries and linear luminaires installed directly above patient beds. They enable the visual assessment of a patient’s health condition and conduct required tests and medical procedures.

## 6. Limitations of the Study

The prepared functional background for temporary hospitals focused on the architecture of COVID-19 temporary hospitals is based on the currently available research materials. Due to their insignificant number, some of the data included in the analyses originate from press releases published in countries where an attempt was made to create field hospitals to combat the effects of the COVID-19 pandemic. Despite the two years of the pandemic, studies on evaluating such facilities are still ongoing. Nonetheless, the conducted analyses of the AmberExpo temporary hospital enables an in-depth study of the topic.

The case study presented in this article, being the only implementation discussed, is focused on a single building type (a temporary hospital in an adapted exhibition and trade fair hall) in one region (Pomorskie province in Poland). This limits the potential of obtaining general conclusions due to homogeneous context related to climate, culture and the situation in the state health care policy. These factors may impact the validity of applying technical solutions, and it is necessary to include other building parameter configurations in different scenarios. For this reason, it is preferable to conduct in-depth analyses of the architecture and infrastructure of other emergency area adaptations completed in other countries. 

Particular care needs to be exercised when generalizing the questionnaire results obtained since it was conducted under restricted conditions due to the ongoing pandemic. The executed questionnaire was based on Google Forms, and due to the epidemiological context, it was impossible to conduct in-depth studies in person. This situation also led to a limitation in the number of questionnaire respondents. 

The analyses in this article apply to architecture and provide only the basic parameters of its supporting construction disciplines that are significant in the article’s perspective. They prove the need for in-depth studies on temporary hospital technical infrastructure in the form of partial analyses focusing on, among others, acoustics, natural or artificial lighting and adaptability in the context of various threats and functionalities.

## Figures and Tables

**Figure 1 ijerph-20-00639-f001:**
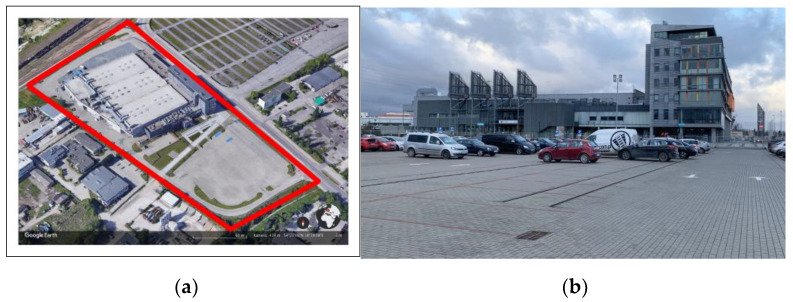
(**a**) Location and development around the Gdansk International Fair building in Gdansk, Poland (source: Google Earth). (**b**) View of the MTG entrance area (photo: Agnieszka Gebczynska-Janowicz).

**Figure 2 ijerph-20-00639-f002:**
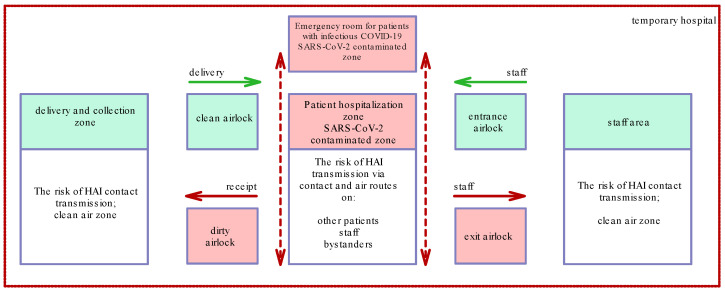
Diagram showing the use of architectural tools for limiting nosocomial infections (author: Rafal Janowicz).

**Figure 3 ijerph-20-00639-f003:**
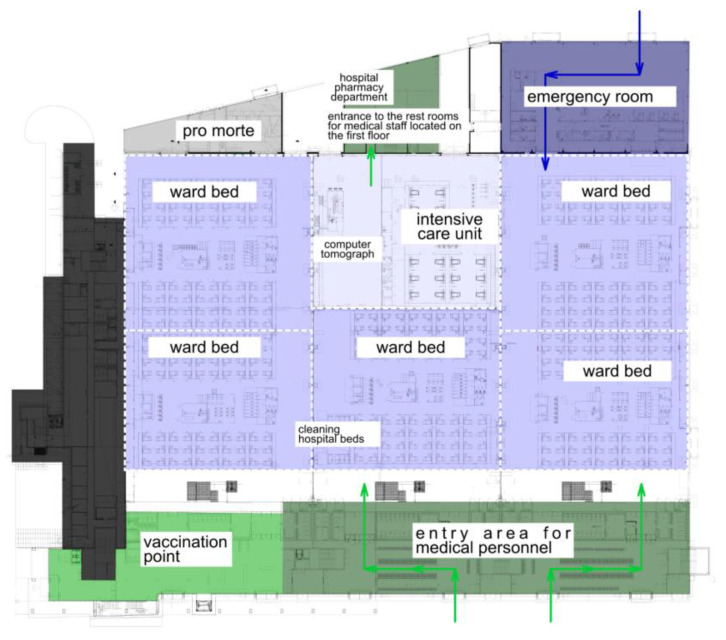
Functional and spatial layout diagram for the AmberExpo temporary hospital in Gdansk (author: Rafal Janowicz): 1. casualty ward; 2. ward bed; 3. intensive care unit; 4. computer tomograph; 5. hospital bed washing station rooms; 6. entrance to the restrooms for medical staff located on the first floor; 7. hospital pharmacy department; 8. mortuary (pro morte); 9. entry area for medical personnel; 10. vaccination point.

**Figure 4 ijerph-20-00639-f004:**
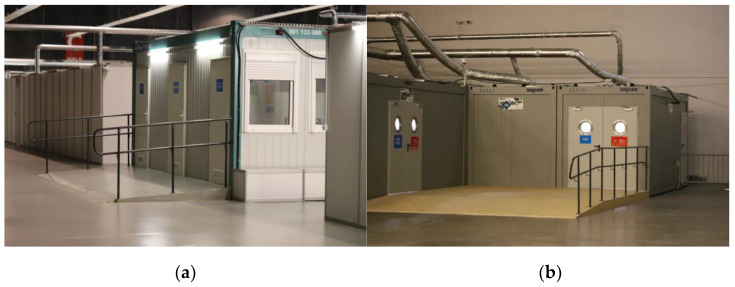
(**a**) Container-type sanitary rooms (e.g., toilets for patients); (**b**) view of the casualty ward; (**c**) top view of the temporary hospital room arrangement and (**d**) view of one of the patient rooms (photo: Agnieszka Gebczynska-Janowicz).

**Table 1 ijerph-20-00639-t001:** Demographic characteristics associated with respondent profession.

Participant Characteristics, Entire Cohort, N = 32	
Gender	
Female	23 (71.9%)
Male	9 (28.1%)
Age, years	
18–29	7 (21.9%)
30–49	16 (50%)
50–60+	9 (28.1%)
Occupation	
Hospital nurses	18 (56.3%)
Hospital physicians	9 (28.1%)
Technical support	4 (12.5%)
Cleaning personnel	1 (3.1%)

**Table 2 ijerph-20-00639-t002:** Questionnaire-based survey related to assessing the technical and organisational solutions at the AmberExpo temporary hospital.

Technical and Organisational Solutions Assessed by the Employees of the MTG Temporary Hospital, Entire Cohort, N = 32	
1. Would you say that the technical and organisational solutions applied in the MTG temporary hospitals made your work environment within the facility safe?	
Yes, definitely	12 (37.5%)
Rather yes	18 (56.3%)
Rather no	1 (3.1%)
No, definitely	1 (3.1%)
Hard to say	0 (0%)
2. How would you rate the functional and spatial solutions in the MTG temporary hospital (staff/material sluice rooms, patient zone partitions) providing the personnel with sanitary and hygienic safety?	
Very Good	10 (31.3%)
Good	18 (56.3%)
Correct	3 (9.4%)
Poor	1 (3.1%)
Very Poor	0 (0%)
Hard to say	0 (0%)
3. How would you rate the need for a special social area for patients within the nursing area?	
Very Good	8 (25%)
Good	12 (37.5%)
Correct	4 (12.5%)
Poor	1 (3.1%)
Very Poor	0 (0%)
Hard to say	7 (21.9%)
4. How would you rate the need for a pneumatic tube system within the MTG temporary hospital?	
Very Good	20 (62.5%)
Good	8 (25%)
Correct	1 (3.1%)
Poor	0 (0%)
Very Poor	0 (0%)
Hard to say	3 (9.4%)
5. How would you rate the need for a bed washing station within the temporary hospital?	
Very Good	8 (25%)
Good	14 (43.8%)
Correct	4 (12.5%)
Poor	0 (0%)
Very Poor	0 (0%)
Hard to say	6 (18.8%)
6. How would you rate the need for an intensive care unit within the MTG temporary hospital?	
Very Good	17 (53.1%)
Good	7 (21.9%)
Correct	4 (12.5%)
Poor	0 (0%)
Very Poor	1 (3.1%)
Hard to say	3 (9.4%)
7. How would you rate the need for an imaging diagnostics area within the temporary hospital?	
Very Good	20 (62.5%)
Good	9 (28.1%)
Correct	1 (3.1%)
Poor	0 (0%)
Very Poor	0 (0%)
Hard to say	2 (6.3%)

**Table 3 ijerph-20-00639-t003:** Questionnaire-based survey related to a comparison between care units for patients infected with or suspected as infected with SARV-COV-2 in the AmberExpo temporary hospital and other medical facilities.

8. Based on your experience from working in other medical facilities, were the working conditions in the MTG temporary hospital significantly different from the working conditions in other wards you previously worked?	
Yes, definitely	24 (75%)
Rather yes	5 (15.6%)
No	3 (9.4%)
9. Comparing various areas of care of patients infected or suspected of being infected with SARS-COV-2, which elements of the technical infrastructure at the MTG temporary hospital would you rate as better solutions than those used in other medical facilities you had worked at?	
Spatial development and communication between units within a facility;	18 (56.3%)
Mechanical ventilation;	8 (25%)
Room temperature;	10 (31.3%)
Facility acoustics;	2 (6.3%)
Artificial lighting;	3 (9.4%)
Medical equipment;	22 (68.8%)
I have no knowledge of this field	4 (12.5%)
Other: Presence of paramedics	1 (3.1%)
10. Which elements of the technical infrastructure at the MTG temporary hospital would you rate as a worse solution than that applied in other medical facilities you had worked at?	
Spatial development and communication between units within a facility;	3 (9.4%)
Mechanical ventilation;	2 (6.3%)
Room temperature;	8 (25 %)
Facility acoustics;	15 (46.9%)
Artificial lighting;	19 (59.4%)
Medical equipment;	2 (6.3%)
I have no knowledge of this field	5 (15.6%)
Other: large distances to cover	1 (3.1%)

## Data Availability

The data presented in this study are available on request from the corresponding author.

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
