# Peer review of "Evaluation of Medical Staff Satisfaction for Workplace Architecture in Temporary COVID-19 Hospital: A Case Study in Gdańsk, Poland"

_ijerph, 2022, doi:10.3390/ijerph20010639_

Round 1

Reviewer 1 Report

The reviewer considers the article presented to be of high quality and congratulates the authors on the presentation of interesting research results. 

I would like to recommend that all the spatial configurations of the individual elements shown in the figures are also described in the text: which functional part should be adjacent to which?. This will allow designers of similar solutions in the future to build on the results that are reported in the reviewed article. 

The reviewer also believes that the authors should provide the name(s) of the design team that created the described facility. In the Polish construction law system, a permit for the construction of a temporary facility has certainly been issued and the name(s) of the designers can be found here: architects, installers, and medical consultants. 

Author Response

Response to Reviewer 1 Comments

Point 1: I would like to recommend that all the spatial configurations of the individual elements shown in the figures are also described in the text: which functional part should be adjacent to which?. This will allow designers of similar solutions in the future to build on the results that are reported in the reviewed article. 

Response 1: Below the graphic diagrams (Fig.2), a paragraph was added explaining the location of the hospital's basic functional zones:

“The functional layout of the temporary hospital can be divided into three main functional interconnected zones: patient care rooms, rooms for medical personnel and a zone of supply and storage of materials necessary for the provision of medical services. Each zone should be separated and communication between them should take place using airlocks and entrances protected by access control. The location of the auxiliary zones should on the one hand be outside the patient care area, but in the immediate vicinity, so as not to cause the need for redundant work resulting from the movement of personnel and materials between clean and contaminated areas.”

Design guidelines in the form of advantages and disadvantages of adapting the exhibition building for the needs of the temporary hospital have also been added to the "Conclusions" section. A similar solution was suggested by the second reviewer.

Point 2: The reviewer also believes that the authors should provide the name(s) of the design team that created the described facility. In the Polish construction law system, a permit for the construction of a temporary facility has certainly been issued and the name(s) of the designers can be found here: architects, installers, and medical consultants.

Response 2: The design teams of the temporary hospital project have been added to the section "General Characteristics of the Study Sites".

Reviewer 2 Report

Dear Authors,

It was a pleasure reviewing your manuscript. Generally, the paper is well-written, however, the methods and results section need to be restructured. The introduction is well-articulated and the problem is well-stated.  Please find below minor comments:

1. You received consent from consent of MTG administrators. However, did you study receive and IRB approval?

2. How did you formulate the survey questions and can you add the questions to the methods section after description of participants.

3. The methods section is fine, some of the sections in the results section should move into the methods section. Also, avoid redundancy in the description of the site and the description of submitting the questionnaires to employees (it was mentioned twice in the paper in two different sections). The results section should only have the results of the surveys as that was the only tool used. 

4. Move the limitation section to after the conclusion section.

5. I noticed that in the results section, you included statements or phrases that should move to the literature review section. One example is when you talk about the benefits of lighting on line 327 -330. This makes the paragraphs representing the findings too long and difficult to kind of quickly understand what you found. 

6. The discussion section could connect what you found in literature review such as (327-330) to the findings.

7. I would try and give a summary in the conclusions section about which architectural elements were the most helpful to healthcare staff during the pandemic. Those could use the language of "design recommendations" 

Good luck to the authors!

Author Response

Response to Reviewer 2 Comments

Point 1: You received consent from consent of MTG administrators. However, did you study receive and IRB approval?

Response 1: We have not applied for IRB approval.

Point 2: How did you formulate the survey questions and can you add the questions to the methods section after description of participants.

Response 2: The questionnaire is comprised of two parts. The first part contains three questions concerning the demographic characteristics of the participants taking part in the study. Specifically, these questions concern gender, age and healthcare profession of these people.  The second part contains 10 questions concerning various aspects of the assessment of satisfaction with architectural and technical solutions adopted at the AmberExpo hospital. Two of them are multiple choice closed-end questions, while the other elements use the answer formula according to the Likert scale - the format of utterances arranged in order from total acceptance to total rejection.

The above text has been added to the article.

Point 3: The methods section is fine, some of the sections in the results section should move into the methods section. Also, avoid redundancy in the description of the site and the description of submitting the questionnaires to employees (it was mentioned twice in the paper in two different sections). The results section should only have the results of the surveys as that was the only tool used. 

Response 3: The description of the architectural and technical solutions of the AmberExpo hospital was included in the article as a "Case study", that is a qualitative study used in the discipline of architecture, therefore it was placed in the Results section. We have changed the order of the description of the research tools "Study location" and "Study Design".

One description of distributing the questionnaires to the employees has been removed - thank you very much for pointing out the repetitions.

Point 4: Move the limitation section to after the conclusion section.

Response 4: The 'Limitations of the Study' section has been moved after the 'Conclusions' section.

Point 5: I noticed that in the results section, you included statements or phrases that should move to the literature review section. One example is when you talk about the benefits of lighting on line 327 -330. This makes the paragraphs representing the findings too long and difficult to kind of quickly understand what you found. 

Point 6: The discussion section could connect what you found in literature review such as (327-330) to the findings.

Response 5-6: Lines 327-330 have been moved to the 'Conclusions' section, where a paragraph has been added about the advantages and disadvantages of adapting exhibition facilities to temporary hospitals.

Point 7: I would try and give a summary in the conclusions section about which architectural elements were the most helpful to healthcare staff during the pandemic. Those could use the language of "design recommendation”.

Response 7: A paragraph with recommended design guidelines was added to the 'Conclusion' section in the form of showing the advantages and disadvantages of adapting the exhibition hall for the needs of a temporary hospital:

The analysis of the adaptation of the construction structure of the exhibition building to the needs of the medical facility made it possible to distinguish the advantages and disadvantages of such a solution. The building described in the article, thanks to the flexibility of the functional system and large area, enables the location of a large number of hospital beds and the optimal distance between the beds, reducing epidemiological risks. Most technical infrastructure, such as mechanical ventilation, electrical network, industrial cameras and fire protection, in this case only need to be redeveloped and adapted to the characteristics of the medical facility. The addition of new infrastructure, such as medical gases, is made possible by the technical ducts located under the floor. The experience of other temporary hospital investment projects shows that the presence of technical ducts is a key asset of a building intended for the temporary performance of medical functions [40]. These aspects mean that in a short time it is possible to plan working conditions similar to those in standard hospitals.

A significant disadvantage of creating a temporary hospital in the exhibition building is the lack or not enough of natural light provided to workstations. Previous research revealed that natural light (daylight) very positively impacts the health and well-being [41] and is of great significance in patient recovery and their contact with the outside world and the day/night cycle change [42]. In contrast, poorly designed artificial night lighting for hospital employees and patients can have an adverse effect [43]. In the case of a patient ward, the light only comes from skylights, is limited and functionally insufficient. For this reason, it was necessary to additionally illuminate the interior of the facility using artificial LED lighting in the form of suspendant luminaries and linear luminaires installed directly above patient beds. They enabled to visually assess a patient’s health condition and conduct required tests and medical procedures.